# Recultivation of Post-Mining Disturbed Land: Review of Content and Comparative Law and Feasibility Study

**Margarita Ignatyeva** [1,2]**, Vera Yurak** [1,2,]*** and Natalia Pustokhina** [3]

[1] Research Laboratory of Disturbed Lands' and Technogenic Objects' Reclamation, Ural State Mining University, 620144 Yekaterinburg, Russia; rinis@mail.ru
[2] Center for Nature Management and Geoecology, Institute of Economics, The Ural Branch of Russian Academy of Sciences, 620014 Yekaterinburg, Russia
[3] Department of Economics and Management, Ural State Mining University, 620144 Yekaterinburg, Russia; ief.etp@ursmu.ru
* Correspondence: vera_yurak@mail.ru; Tel.: +7-343-278-7385

**Abstract:** The article considers the concept of the circular economy as an important tool for achieving sustainable development, which relates to the preservation of renewable resources' mass through the renewal of withdrawn resources and the restoration of disturbed ones. It is directly linked to remedial land treatment in post-mining disturbed land. However, after numerous studies, the conceptual apparatus of recultivation remains unspecified. Moreover, there is a gap regarding the trends of evolutionary changes in studies of legislation and feasibility on the subject of recultivation. Employing comparative law as a tool, the aim of the study is to develop a consistent approach based on circular economy by establishing the stages of legal support for recultivation and identifying the content of all these stages regarding economic efficiency. Currently, the environmental priorities of the economy are triggering the usage of the ecosystem approach for assessing the ecological result of recultivation. Therefore, the core of the paper is the development of a consistent circular economy approach by (1) clarifying the concept of recultivation, (2) identifying the stages of the development of a legal framework for recultivation and (3) revealing evolutionary changes in feasibility studies on recultivation. The authors prove that recultivation should be considered from the perspective of geoaesthetics, which implies a harmonious incorporation of the recultivated landscape into the environment.

**Keywords:** mining; circular economy; sustainable development; evolutionary change; revitalization; renaturation; restoration; environmental remediation; rehabilitation; reclamation

## 1. Introduction

Any human activity is associated with impact on the natural environment, which includes (1) the seizure of natural resources, (2) environmental pollution and (3) surface disturbance. Every year, man-made pressure on nature increases. The amount of disturbed land in the world is increasing gradually [1,2]. Furthermore, over 3.5 million perceived brownfield sites in North America and Europe remain ignored [3]. In the case of Russia, more than 30 million tons of pollutants enter the atmospheric air annually, and 19% of wastewater is discharged into water bodies without treatment. In almost all regions, soils deteriorate due to water and wind erosion, land flooding and waterlogging. Desertification has affected more than 100 million hectares; another 18 million hectares are polluted soil zones, located around industrial complexes. About 4 billion tons of production and consumption waste are generated annually, while about 4 million hectares are occupied by authorized waste disposal

facilities. More than 30 million tons of production and consumption waste have been accumulated, of which more than 400 thousand tons are highly toxic. The amount of waste not involved in secondary economic turnover is growing [4–6]. According to expert estimates, the annual loss of Russia's GDP due to environmental degradation (excluding damage to human health) is from 4% to 6% (Strategy for environmental safety of the Russian Federation for the period until 2025, approved by Decree of the President of the Russian Federation on 04/19/2017 No. 176). The mining complex plays an important role in the deterioration of environmental safety [7]. While it does not have the highest damage intensity among industrial production activities [8–13], it affects all elements of the biosphere [14–18]. Moreover, it is a cause of excavation and movement of huge masses of the lithosphere that also lead to the loss of a range of ecosystem services [15,19]. Research [20] shows that there are between 1.1 and 6.7 units of waste per unit of solid mineral extracted in mining. These units of waste are usually located on the surface afterwards. As a result, areas of disturbed land are appearing where the bulk consists of dumps and tailings (in iron ore subsoil uses, these occupy from 62% to 75% of land allotment; in copper ore subsoil use, they occupy even more) [20]. The Russian case study indicates that the regions with a high level of land degradation are the Ural, Siberian and Far Eastern regions. These three regions form the mineral resource centers of the Russian Federation. In all disturbed land, violations related to the development of mineral deposits are common, making up about 80% [21]. The largest share of the disturbed land is taken by open-pit mining. Violations also pertain to the lithosphere, where the formation of man-made voids occurs with and without access to the surface. In the former case, these voids result from quarry excavations of existing and working open pits and the collapse zones of existing and closed mines. In the latter case, they are underground technogenic voids. It is undeniable that the rate of anthropogenic-caused environmental change is much higher than the rate of ecological balance restoration [22], triggering the need for the timely restoration of disturbed land and the lithosphere. This fact led to the creation of recultivation treatment. Various land conservation activities have been observed and recorded through the centuries. The Phoenicians, Romans, Chinese, Incas, and Mayans constructed erosion-control terraces or contoured their fields [23]. Nowadays, according to J.C.S. Rosa et al. [14] (p. 1), "Mining companies are usually required to prepare a rehabilitation plan in cooperation with other stakeholders by considering a set of variables and criteria, such as groundwater level, final pit and waste dump landforms, stabilisation and revegetation strategies, land tenure, land use regulations, and interests of stakeholders, as well as the pre-mining environment" [24,25]. The content of recultivation is largely explained in existing research [26], where authors defined three categories of remedial land treatment according to the USA National Academy of Sciences [27]: rehabilitation, reclamation and restoration. T.J. Terrence [26] also defined the term "reclamation" and mentioned some legal pluralism in the understanding of recultivation: "The terms rehabilitation, reclamation, and restoration have not been used consistently and connotations have varied through the years. The pertinent laws and regulations have been interpreted and enforced in different ways from time to time and from place to place". A.M. Gaidin [28] used the term "revitalization" for remedial land treatment, which is very similar to the USA National Academy of Sciences' term restoration. E. Kalita and J. Baruah [29] used the term "environmental remediation". According to them, "remediation and reclamation of polluted environments are among the biggest challenges faced by the global community toward providing sustainable living conditions for the generations to come". The US Market Research Report, 2013, stated that the remediation and environmental cleanup services industry had grown by 1.6% over the past five years and was poised to be a $122.8 billion industry by 2022 [30]. Therefore, the recultivation market is a highly developed one [31]. However, despite numerous studies [26–29,32–45] and in view of all the above information, a certain definition of the conceptual apparatus of recultivation remains unspecified. Moreover, there is a gap in the trends causing evolutionary changes in recultivation legislation [26,37,40,46–62] as well as in the economic justification for setting up recultivation projects [63–102]. Therefore, the core of this paper is the development of the theoretical and methodological foundations of the circular economy by (1) clarifying the concept of recultivation, (2) identifying the stages of development of the legal

framework governing the restoration of disturbed land and (3) revealing the evolutionary changes in feasibility studies on recultivation, largely due to changes in ecological and economic models and in the "society–nature" system.

## 2. Materials and Methods

The research framework included three steps of analysis. In individual steps, various methods and materials were used, which are summarized below:

*Step 1.* Clarification of the concept of recultivation.

Applying a systemic and evolutionary approach based on at least a hundred papers [1–102] to the task, the concept of recultivation was clarified (see Section 3.1). We prove that recultivation should be considered from the perspective of geoaesthetics in regions with a high concentration of population. This implies a harmonious incorporation of the recultivated landscape into the environment.

*Step 2.* Identification of the stages of the development of the legal framework governing the restoration of disturbed land.

Legal support of recultivation in European countries, the former USSR, the USA and the Russian Federation, as well as scientific research, was used to clarify the concept of recultivation (see Section 3.2). For conducting the research, the authors used databases from the USA, such as American State Papers, 1789–1838, the Avalon Project database, Congress.gov and Guide to Law Online. In relation to European countries, we also used the Foreign and International Law Resources Database (HeinOnline Databases), and we used Consultant+ for the former USSR and the Russian Federation. By using comparative law as a tool, the legal framework of recultivation treatment was considered. The stages of recultivation's evolution as a legal institution and the basic obstacles limiting the elaboration of this institution as well as ways of removing these restrictions were identified.

*Step 3.* Revelation of the evolutionary changes in feasibility studies on recultivation.

Regarding the evolution of feasibility studies on recultivation, our analysis of the studies focused on identifying the historically established methods and tools for assessing the specific effects of both engineering and the stages of biological recultivation. For preventing economic damage from disruption to regulation and the flow of cultural (social) ecosystem services, an ecosystem services approach was implemented. Therefore, in the context of different climatic zones, the "regulating soil erosion" ecoservice was evaluated up to the date 01/01/2020, based on the Scopus database and Environmental Valuation Reference Inventory (EVRI) data (see Section 3.3).

The research hypothesis was that the development of a consistent circular economy-based approach could contribute to the move to sustainable development and to the achievement of sustainable development goals by 2030. It is quite interesting that the origins of the idea of sustainability, in the German expression *Nachhaltigkeit*, lie in the 18th century [32,33]. The development of a consistent circular economy approach was made by (1) clarifying the concept of recultivation; (2) identifying the stages of the development of the legal framework governing the restoration of disturbed land; and (3) revealing the evolutionary changes in feasibility studies on recultivation, largely due to changes in ecological and economic models and in the "society–nature" system. The research was based on the systemic, evolutionary and ecosystem approaches. A meta-analysis was carried out and indicated significant differences in the understanding of post-mining development in different countries, caused by many aspects, such as the ecological engineering of landscapes, responsibility and property rights issues and land rarity criteria. Legal support for recultivation in European countries, the former USSR, the USA and the Russian Federation, as well as scientific research and databases relating to ecosystem valuation formed the information base of this paper. The authors' observation period is from the 18th century to the present.

## 3. Results and Discussion

*3.1. Concept of "Recultivation": The Content Transformation from Ordinary "Cultivation" to "Revitalization" or "Renaturation", or "Restoration", or Even "Environmental Remediation"*

The term "recultivation" first appeared at the beginning of the 20th century. However, the first attempts to restore disturbed land in Germany dated back to the end of the 19th century. In the USA in 1937, massive greening of worked-out areas of coal quarries was carried out. In Germany, as early as 1923, 242 hectares of dumped lands were afforested in the brown coal basin [34]. Initially, as the Russian and foreign experience shows, land recultivation was defined as a process where "a complex of various works (engineering, mining, reclamation, agricultural, forestry, etc.) are carried out over a certain period and aimed to restore the productivity of disturbed territories and to return them to different types of use" [34,35]. It was understood as "a special soils' restoration procedure for agricultural or field use" by V. Lazareva, who highlighted the foreign experience of recultivation, and first used this term in her scientific work [36]. To some extent, it corresponds with the well-known concept of "cultivation", coming closer to the stage of biological recultivation. This also reflects to the understanding of recultivation used by V.V. Tarchevsky and E.M. Lavrenko ( "industrial botany", "industrial biogeocenology") [37]. Criticizing one-sided views of recultivation, in which it is considered only as "the procedure that refers to the return of post-mining disturbed lands to economy and rational use" [38,39], the authors of [40] (p. 11) believed that "recultivation is a process aimed not only to partially transform post-mining disturbed lands, but also to create even more productive and rationally organized territory included in the cultural anthropogenic landscapes. Therefore, it is the optimization of technogenic landscape and the improvement of environmental condition". However, employing an aim-oriented approach, in 1974, the National Academy of Sciences of the USA defined in their research three categories of remedial land treatment and stated that industry favors rehabilitation, regulatory authorities favor reclamation and many ecologists favor restoration. In rehabilitation, "The land is returned to a form and productivity in conformity with a prior land-use plan including a stable ecological state that does not contribute substantially to environmental deterioration and is consistent with surrounding aesthetic values. Rehabilitation usually permits the greatest flexibility in future land use and incurs the least cost" [27]. In reclamation "The site is hospitable to organisms that were originally present or others that approximate the original inhabitants. Reclamation infers that the pre- and post-disturbance land uses are nearly the same" [27]. In restoration, "The condition of the site at the time of disturbance is replicated after the action. Restoration allows no land-use flexibility and incurs the greatest cost" [27]. Within the framework of this aim-oriented approach, in 1998, J.T. Terrence defined reclamation as "the treatment of disturbed areas to create stable landforms and edaphic conditions to sustain predetermined land uses with minimal maintenance" [26] (p. 4018).

The understanding of recultivation understanding has changed in the 21st century. The term "recultivation" is increasingly replaced by the terms "revitalization", "renaturation" or "restoration" (according to the classification of the National Academy of Sciences of the USA), i.e., it is the creation of an updated landscape [41–44], a landscape of high aesthetic value, which requires the involvement of landscape designers. According to research [45], the updated landscape should satisfy the following requirements:

- "to be environmentally friendly;
- to harmonize with the natural environment, complementing the missing elements, increasing the number and variety of ecological niches;
- to meet aesthetic requirements;
- to meet the present and future needs of the local population and the region".

According to the 21st century understanding of recultivation and the circular economy and sustainable development trend, in 2020, E. Kalita and J. Baruah stated that "Environmental remediation

refers to the reduction/removal of pollutants or contaminants from water and soil for the protection of living systems and the environment against further deterioration for a sustainable future" [29] (p. 525).

*3.2. Legal Support for Recultivation Treatment*

*First stage.* It is believed that the first law relating to the restoration of disturbed land was adopted in the United States (state of West Virginia, 1939), when the concept of disturbed land and the associated adverse effects became extremely tangible. In 1940, Germany passed directives for the restoration of open spaces caused by open-cast mining. The main requirements for restoration were: "(1) the removal and application of soil cover; (2) disposal or neutralization of toxic waste; (3) elimination of the worked-out space of coal quarries and mining workings; (4) restoration of the original terrain; (5) restoration of vegetation" [46] (p. 7). Almost no attempts at this had been made in the USSR at this time. This was the initial stage in the development of recultivation legislation. This stage can be framed as 1930 to 1955, when the development of recultivation legislation was characterized by low activity.

*Second stage.* During the postwar period (1955–1974), there was a sharp surge in legislative activity. Laws were created requiring recultivation, and international and national conferences and symposia devoted to recultivation intensified. This period could be called the main one when a basis for recultivation legislation was created. Recultivation became a part of the general planning of conservation and landscape development, referred to as "landscape recultivation" [47]. The United States passed many laws on recultivation in almost all states in 1953–1963. There were different units on recultivation in every state, such as mine bureaus, forest departments, agricultural departments, recultivation departments, etc. [48]. In England, in 1958, a law on open-pit coal mining was adopted, which included a requirement for the recultivation of post-mining disturbed land. In Germany, in the Ruhr basin, recultivation was carried out since 1956 according to plans that were developed simultaneously with mining plans. After the end of the Second World War in the German Federal Republic, laws were adopted in most regions to protect landscapes from destruction caused by open-cast mining. In addition to the technical aspect of nature conservation (the recultivation of disturbed land), much attention was also paid to the conservation aspect (the preservation of undisturbed landscapes) [40]. In the Soviet Union during the 1960s, many attempts were made at setting up a recultivation institute. This fact is proved in [49–52] and other papers as well as statement from conferences and research teams [53–56], etc. The first documents [37] were developed from the late 1960s to the 1980s, mainly by specialists of the Ministry of Agriculture of the USSR and its subordinate institutions [57]. However, legislative acts relating to the restoration of disturbed land continued to be absent, except the fact that subsoil users had to restore disturbed land to a suitable state for economic use. This requirement was mentioned in republican laws on nature protection (1957–1963) and in the USSR and associated republics' Fundamentals of land legislation (1968). This means that the main stage for the USSR was shifted to 10–15 years later. Recultivation in Russia is governed by national standard 17.5.2.02-83, "Classification of disturbed lands for recultivation considering subsequent use". Recultivation requirements are governed by national standard 17.5.3.04-83, "General requirements for land recultivation".

*Third stage.* The legislative basis created during the second stage predetermined the formation of the next organizational stage, during which (1) infrastructure had to be created to ensure the establishment of recultivation treatment and (2) an economic mechanism was developed. This mechanism was intended to be aimed at stimulating recultivation treatment. All European countries and the USA are still on this stage today. These countries have achieved an excellent result in organizational and economic infrastructure development, and they are still improving this as well as its legislative base. However, a German case study showed that this country still has problems with its mechanism and infrastructure [58]. For instance, the Mansfeld district, which is associated with a more than 1000-year-old copper-mining tradition, was included in the German federal program "Ökologische Großprojekte" (Major Ecological Projects) for recultivation treatment [59]. Another similar example is the coal-mining region Zwickau-Lugau-Oelsnitz. However, "one of the most crucial problems for

the Zwickau-Lugau-Oelsnitz is the lack of outside funding for the new phase of rehabilitation" [60]. These examples indicate serious obstacles to the economic mechanism of recultivation treatment in Germany. As far as both infrastructure and this economic mechanism are conserned, S. Krøijer and M. Kollöffel demonstrated that protests have been raised around mining issues in the country in research [61]. Looking at the Russian tradition, it was unfortunately the time-consuming transition to the market economy for Russia that caused the recultivation issue to become secondary for decades.

*Fourth stage, for the Russian case study only.* A surge of attention to the recultivation of disturbed land was observed in recent years only. This surge was linked to the continuous ecological deterioration of the environment and the appearance of governmental management aimed at preventing these ecological obstacles. It trigged the spread of new economic models and paradigms, such as the "green economy" and circular economy, that focused primarily on resource conservation. However, despite the huge attention to recultivation even in Russia, legislation deficiencies still exist: (1) there is no legislatively fixed composition and content for project documentation for recultivation, including engineering surveys; (2) the legislation does not establish the control of disturbed land after the recultivation project has been completed; (3) there are contradictions in the legal acts regulating the process of coordinating the projects; (4) sanctions in response to failure to fulfill recultivation obligations have low validity; (5) there is a lack of legislation for the creation of a database of disturbed land, including lands on which "ownerless" wastes are disposed [62,63].

The time frame of the evolution of legal support for recultivation treatment is presented in Figure 1.

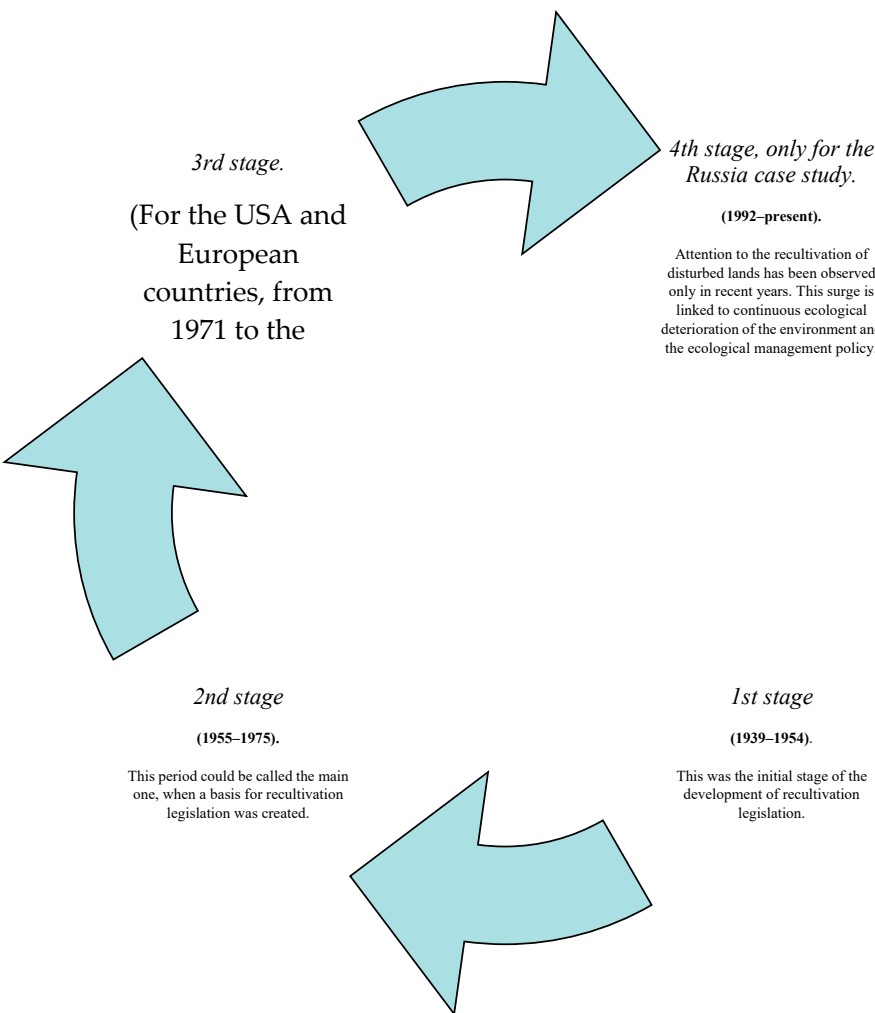

**Figure 1.** Time frame of the evolution of legal support for recultivation treatment.

### 3.3. Evolution in Feasibility Studies of Recultivation Treatment

The accumulated world experience of recultivation treatment indicates the importance of economic feasibility studies [64–66] that help to choose the most effective and economical ways to return all disturbed land to secondary economic use [48] (p. 78).

In the first stage, the economic efficiency of recultivation was linked to the value of the disturbed land, considering the agricultural methods used for recultivation.

In order to carry out calculations, the All-Union Research Institute of Agricultural Economics (USSR) carried out preliminary evaluation of the average price of 1 hectare of agricultural land in the republics. The basis of the calculation was the net income of collective farms from 1 ha of agricultural land:

$$Pr = (NI/r) \times 100 \tag{1}$$

where $Pr$—price of 1 ha of agricultural land, c.u. (c.u.—currency unit) per ha; $NI$—net income from 1 ha of agricultural land, c.u. per ha; $r$—loan interest, %.

In cases where the qualitative parameters of agricultural land were different from the average values, it was recommended to use the grade-rating method:

$$Pr_1 = (Pr_{av} \times m_1)/m_{av} \tag{2}$$

where $Pr_1$—price of 1 ha of land of a certain quality, c.u. per ha; $Pr_{av}$—average price of 1 ha of agricultural land, c.u. per ha; $m_1$—land valuation by net income (of the given land plot), grade; $m_{av}$—total estimated grade based on the net income of agricultural land, grade.

The average price for 1 ha in the USSR was about RUB 309, in the Moldavian and Ukrainian SSRs it amounted to RUB 1780–1024, in the Kazakh SSR it was RUB 112, and in the Russian Soviet Federated Socialist Republic (RSFSR) and the Lithuanian SSR it was RUB 320–327 [49]. The value of 1 ruble (RUB) varied from USD 0.6 to 1.85 from the 1960s to the 1990s.

When determining the payback period for recultivation treatment, the recultivation costs were compared with the possible annual income from agricultural products received from the restored territory with or without a time factor [67,68]. Recultivation costs were limited by the standard cost of a new land development in return for land withdrawn for non-agricultural needs [69,70]. The first enforcement demands for compensation for alienated lands date back to 1939, when the Hydropower Project issued the "Instructions for damage valuation of flooding in the design of hydraulic structures", providing a calculation of monetary compensation based on the principle of "hectare per hectare" (the cost of developing an equal amount of hectares regardless of their productivity). The same principle was used for compensation in a number of union republics, approved the 1960s. In 1976–1978, standards for compensation for land allotment were introduced, in which, in addition to the costs required for the development of an equal area, the need for land reclamation measures was also taken into account. As a result, production on these lands would correspond to the volume of production on the lands previously withdrawn [71].

For example, according to the RSFSR, the standards for the development of new lands instead of withdrawn ones for non-agricultural purposes amounted to RUB 5160–9160 per ha for arable land and RUB 3120–4990 per ha for hayfields and pastures. The differentiation of these standards was determined by the quality of agricultural land, which was measured in grades. In the same period, the research institute for pricing issued guidelines on the inclusion of the recultivation costs of disturbed land in the cost and wholesale prices of non-metallic building materials [72].

In other cases, the agricultural method of recultivation was considered ineffective and other ways of recultivation were required.

According to existing research [40], the economic cost-effectiveness of recultivation should meet the conditions:

$$E_r = \Delta P/(C \times (1 + E_c)^t) \geq 0.06 \tag{3}$$

where $E_r$—coefficient of economic cost-effectiveness of recultivation, unit fraction; $\Delta P$—profit from the sale of products derived from recultivated land, c.u. per year; $C$—recultivation costs, c.u. per year; $E_c$—the standard for bringing different costs at the same time, equal to 0.08; t—biological stage of recultivation, years.

The standard of economic efficiency (profit margin) for the national economy during this period was 0.12.

However, the specifics of recultivation treatment are (1) the duration of the biological process is about 5–10 years after mining completion and (2) the time until an enterprise will use recultivated lands also needs time to reach its design capacity. These specifics predetermined the use of an $E_r$ equal to 0.6 and a payback period of 17 years [40] (p. 227).

The researchers subsequently focused on the details of calculating the amount of costs and income/benefits received from the use of recultivated land, primarily costs. To this end, the authors [73] proposed an expanded structure of costs and effects according to type of recultivation. In addition to the direct costs of recultivation, it was proposed that costs related to the technological processes of mining should be considered. V.D. Gorlov considered that it was necessary to include in the costs "(1) the damage from the violation and loss of the soil layer as the main means of agricultural production; (2) the costs of restoring the former fertility of displaced soils; (3) the damage from the reduction in gross production of agricultural products due to the land transfer to mining allotment" [74] (p. 14).

The author also believed [53] (p. 14) that damage largely depended on the loss of the soil layer. To reduce this loss it is very important to conduct mining with the preservation of fertile soil. Maximally clean excavation of the soil layer, without clogging with rock and waste material, causes little economic damage. The economic damage from the loss of agricultural land, taking into account losses of the soil layer, is evaluated as:

$$CL = S \times (Pr_{av} - (Pr_s \times C_{cs})) \tag{4}$$

where $CL$—the cost of land disturbed by mining, c.u.; $S$—mining area, ha; $Pr_{av}$—average price of 1 ha of agricultural land, c.u. per ha; $Pr_s$—price of the soil layer, c.u. per ha; $C_{cs}$—soil preservation coefficient.

The soil preservation coefficient is evaluated by using the total losses and dilution of the soil layer ($\Sigma LD$, %) according to Formula (5).

$$C_{cs} = (1 - (\Sigma LD/100)) \tag{5}$$

The soil layer's price per hectare can be evaluated from the costs of restoring the fertility of displaced soils or the costs of creating new soils equal in fertility by the formula:

$$Pr_s = (p_1 \times R_{cy}) + (p_2 \times C_{ct} \times B_{cy}) \tag{6}$$

where $p_1$—the full period of restoration (creation) of the structure and fertility of displaced soils, depending on the purity and completeness of their excavation, years; $R_{cy}$—annual restoration costs, c.u. per ha; $p_2$—biological recultivation period, years; $C_{ct}$—cost factor depending on losses and dilution of displaced soils; $B_{cy}$ – annual costs of biological improvement, c.u. per ha.

The research [52] shows that the indicators of payment for land and recultivation treatment per 1 ton of ore, depending on the conservation of soil fertility, differ significantly: at 100% preservation it is RUB 0.089 and at 50% preservation it is RUB 0.123; without preserving fertile soils it is RUB 0.149.

By specifying the content of recultivation costs, E.P. Doronenko indicated that such costs as (1) engineering and biological stages, (2) design and survey work determining the methods of land usage and (3) design and research work on recultivation must be included [75]. The detailed components of the engineering stage, which should be presented in technical projects, are given in [76,77].

As a result, the total costs of the engineering stage ($C_{eng}$, c.u.) include [52]:

$$C_{eng} = C_{rs} + C_{lrs} + C_t + C_{p1} + C_{p2} + C_{ft1} + C_{ft2} + C_c + C_r \tag{7}$$

where $C_{rs}$—the cost of removing rich soil (chernozem), c.u.; $C_{lrs}$—the cost of loading rich soil into vehicles, c.u.; $C_t$—the transportation cost, c.u.; $C_{p1}$—the planning cost for the recultivated dump surface, c.u.; $C_{p2}$—the planning cost of laying rich soil on the surface of the dump, c.u.; $C_{ft1}$—the cost for flattening and terracing the dump's slopes, c.u.; $C_{ft2}$—the planning cost of laying rich soil on the slopes and terraces of the dump, c.u.; $C_c$—the chemical reclamation cost, c.u.; $C_r$—the construction cost for access roads to the land being recultivated, c.u.

Much less elaboration has been done on the emerging income than on the costs. However, the emerging income has gained enough attention from the 1980s to the present. The object of such research has primarily been the calculation of the amount of prevented damage. The structure of the overall effect of recultivation (R) was presented in [73] and included:

$$R = R_{ecol} + R_{econ} + R_{adecon} + R_{soc} = R_{economic} + R_{ecological\&social} \tag{8}$$

where $R_{ecol}$—ecological effect of environmental protection, which implies the creation of normal aesthetic and sanitary conditions; $R_{econ}$—effect in the form of products, obtained in the restored area, or product growth in adjacent areas; $R_{adecon}$—additional effect obtained from the use of overburden rocks; $R_{soc}$—social effect obtained using recultivated areas for recreational purposes.

However, as the researchers admitted, it was impossible to calculate many social factors due to the lack of criteria for evaluating them. The interpretation of the components was initially presented in the draft Guidelines for Determining the Economic Efficiency of Disturbed Lands' Recultivation (1983), and then in the cross-industry Guidelines for Determining the Economic Efficiency of Disturbed Lands' Recultivation (1986) approved by the Deputy Chairman of the USSR State Planning Committee. According to these two guidelines, economic efficiency can be calculated for a one or several methods of recultivation, considering the national economic result or the economic result obtained by an individual enterprise, association, industrial unit, etc. The components of the effect of recultivation effect were specified during the period from the 1980s to the 1990s. The guidelines show that the calculation of the overall effect of recultivation effect consists of two components: (1) the economic outcome of recultivation works ($R_{economic}$), which is composed of $R_{econ}$ and $R_{adecon}$; and (2) the socio-environmental outcome of recultivation works ($R_{ecological\&social}$), composed of $R_{ecol}$ and $R_{soc}$. In turn, $R_{ecol}$ represents an environmental outcome, and $R_{soc}$ represents a nature restoration outcome.

The calculation of the economic outcome of recultivation works does not cause any difficulties, which cannot be said about the socio-environmental outcome of recultivation works. The problem of environmental degradation associated with pollution and land deterioration predetermined the emergence of studies aiming to assess the economic value of natural resources [78,79] and to prevent economic damage caused by ecological problems that had an influenced on human welfare and wellbeing. This damage is the environmental outcome. To calculate it, the guidelines [80] can be used. However, there are no recommendations for calculating the nature restoration outcome, which society can benefit from as a result of the improvement of sanitary-hygienic, recreational and aesthetic conditions. The guidelines [81] (p. 30) state that "it should be considered as the coefficient (ratio) till the time when the experience in calculating this value will be accumulated". In practice, it is a matter of preventing economic damage caused by disruption in regulation and the flow of cultural (social) ecosystem services.

Summing up the Russian case study, we can identify five types of main criteria in feasibility studies of recultivation treatment: (1) benefit type; (2) standard type; (3) cost type; (4) biological type and (5) ecosystem services type. All these types can be used both at the project step of recultivation treatment and at the result step, when recultivation has been done, in order to evaluate the economic efficiency of the treatment. According to scientometric analysis [82], the USA, England, Canada, Germany, China, Italy and the Czech Republic focus on recultivation issues and have published many articles due to land and environment policies. Moreover, "sustainable regeneration, urban brownfields' regeneration, mental distribution, coal-mine brownfield, and ecosystem service were the identified co-citation clusters and represented the hot topics and emerging trends" [82]. As far as the remaining

papers are concerned, for example, a German study [83] aimed at the biological type of feasibility study, where measurement results were compared with previous data from microbial communities in sandy substrates and their impact on the environment. The same type of main criteria were chosen for a feasibility study by Z. Zhang et al. [84]. Cost-benefit types are discussed in [85]. Other observed research concentrates on the ecosystem services type [86–88], etc. Therefore, this review of feasibility studies of recultivation treatment shows the emerging and growing trend of the ecosystem services type during the last years. The main question that arises is that of how to calculate these ecosystem services in order to evaluate the economic efficiency of recultivation treatment.

Among the first Russian researchers [89–92] who worked on the economic evaluation of ecosystem services, there are A.A. Tishkov [90], S.N. Bobylev [91], etc. Some practical examples of their assessments relate to Moscow, Tomsk, the Moscow region, and several state reserves. Even earlier than Russian researchers, foreign authors investigated this issue. The employment of the ecosystem approach made it possible to calculate some regulating and cultural (social) ecosystem services provided by forest, steppe, wetland and other ecosystems, which served as the basis for evaluating economic damage. It became possible to switch from correction coefficients (ratios) to income calculations, which represented the amount of preventable economic damage to obtain due to the preservation of the flow of regulation and of cultural (social) ecosystem services. While the economic assessment's guidelines for social ecosystem services are still at the initial stage of development, guidelines surrounding most regulatory ecosystem services have been successfully developed and tested at several sites. Moreover, these have already been recommended for practical implementation. For the purposes of recultivation and from the point of view of economic evaluation, the most interesting ecosystem service is regulation, such as "regulation of soil erosion" [93–102].

According to the results of our express analysis, the calculation of the "regulation of soil erosion" ecoservice in the context of different climatic zones has usually been done by employing the following methods (Figure 2): (1) market price (64%); (2) willingness to pay (27%); and (3) quantitative evaluation (analogy method) (9%) (The numbers have been calculated by employing the data presented in Table 1). The data of the economic assessment of the ecosystem service "regulation of soil erosion" are presented in Table 1. Economic assessments obtained by previous research [93–102] were recalculated by the authors of this article up to the date 01/01/2020 using a discounting tool. The rate of return was defined as the average refinancing rate of the Central Bank of the Russian Federation for each year. According to Table 1, the value of "soil ecosystems" varied from USD 5.97 to 269054.49 per hectare per year up to the date 01/01/2020.

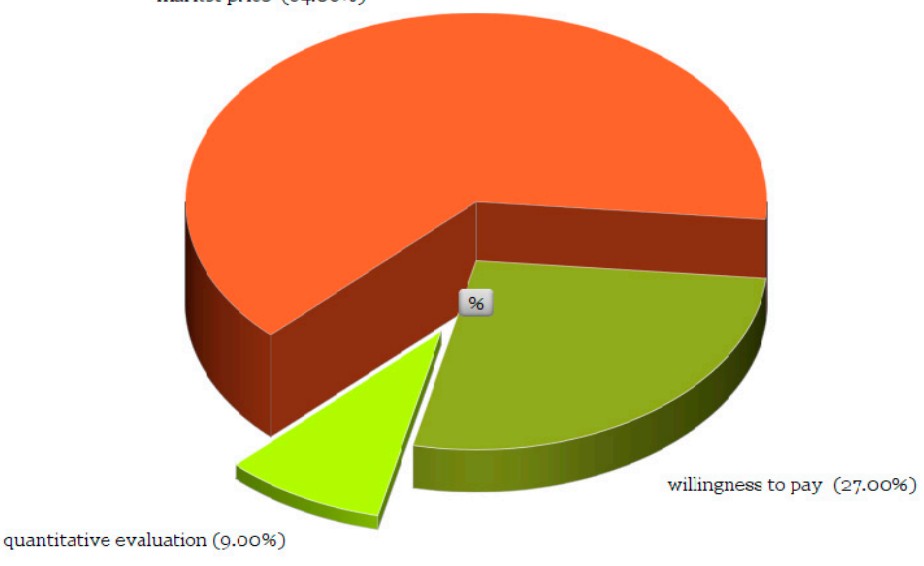

**Figure 2.** Valuation methods for the "regulation of soil erosion" ecosystem service.

**Table 1.** Economic evaluation of "regulation of soil erosion" ecosystem service in the context of the temperate climatic zone.

| Type of Soil | Evaluation Method | Economic Evaluation | Unit | Area, ha | Economic Evaluation | Unit | Convertible Economic Evaluation by 2020 Using the Discounting Tool | Unit | Year of Evaluation | Country (Region/Entity/District/City) | Source |
|---|---|---|---|---|---|---|---|---|---|---|---|
| | | | | | **Temperate climatic zone** | | | | | | |
| podzolic and sod-podzolic | market price | 182.00 | million USD per year | 1022 | 178,082.19 | USD per year per ha | 269,054.49 | USD per year per ha | 2015 | Great Britain, Wales | [100] |
| river floodplains | market price | 63.00 | million USD per year | 987 | 63,829.79 | USD per year per ha | 93,639.05 | USD per year per ha | 2007 | China, Chongqing | [101] |
| gray and brown forest | market price | 1.064 | million USD per year | 785 | 1355.41 | USD per year per ha | 1988.40 | USD per year per ha | 2007 | USA | [102] |
| | | | | | **Subtropical climatic zone** | | | | | | |
| brown and taupe | quantitative evaluation | 17.70 | million USD per year | 1646 | 10,753.34 | USD per year per ha | 10,879.41 | USD per year per ha | 2011 | Spain, Andalusia | [94] |
| red earth and yellow earth | market price | 0.31 | million USD per year | 459 | 675.38 | USD per year per ha | 578.05 | USD per year per ha | 2013 | Japan | [95] |
| red earth and yellow earth | market price | 316.10 | million USD per year | 1756 | 180,011.39 | USD per year per ha | 154,068.96 | USD per year per ha | 2013 | Japan | [95] |
| | | | | | **Subequatorial climatic zone** | | | | | | |
| red earth and yellow earth | willingness to pay | 8.64 | million USD per year | 712 | 12,134.83 | USD per year per ha | 12,873.59 | USD per year per ha | 2018 | Vietnam | [96] |
| reddish brown | market price | 2.30–2.70 | million USD per year | 1345 | 1858.74 | USD per year per ha | 1858.61 | USD per year per ha | 2020 | South and Central Ethiopia, Tabota Coromo, Coromo Danshe | [97] |
| red | market price | 53.70 | million USD per year | 1313 | 40,898.71 | USD per year per ha | 47,415.72 | USD per year per ha | 2019 | Benin, Sacabansi | [98] |
| reddish brown | willingness to pay | 74.4685 | million USD per year | 340,000 | 219.025 | USD per year per ha | 384.67 | USD per year per ha | 2016 | Kenya, Naivasha | [99] |
| | | | | | **Tropical climatic zone** | | | | | | |
| semi-deserts and deserts | willingness to pay | 0.7344 | million USD per year | 150,000 | 490 | USD per year per ha | 597 | USD per year per ha | 2017 | Tunisia | [93] |

Analyzing the maximum limits of economic assessments of the value of the "regulation of soil erosion" ecosystem service demonstrated that the soils of the temperate climatic zone are the most valuable. Their maximum was USD 269,054.49 per hectare per year (Table 1). In second place, there are the subtropical zone soils with a maximum economic value of USD 154,068.96 per hectare per year. In third place, there are the soils of the subequatorial belt. In fourth place, there are the soils of the tropical zone. For the tropical belt, the assessment is not entirely representative, since only one study was found on the assessment of the "regulation of soil erosion" ecosystem service within this belt.

Summarizing all the above information, it should be strongly underlined that the recultivation issue is on the agenda all over the world. One of the reasons is that subsoil use is one of the most significant disruptions to the environment over the last centuries. Moreover, the growth of open mining has led to large-scale violations of land resources.

## 4. Conclusions

Considering the concept of recultivation, the analysis of researchers' views on the concept of "recultivation" shows that it is increasingly being replaced by the terms "revitalization", "renaturation", "restoration" or even "environmental remediation". Such understanding provides the harmonious inclusion of the restored landscape into the environment, taking into account landscape architecture. Identifying the stages of the development of the legal framework governing the restoration of disturbed land, the revealed stage-by-stage evolution of the legislative base of recultivation indicates a 10–15-year quality gap between the Russian legislative framework governing recultivation and the foreign ones. The first experience of landscaping dumps dated to the 1920s–1930s, and the first legislative act relating to the recultivation of disturbed land was adopted in 1939 (USA). The authors identified three stages in the evolution of legal support for recultivation. The first stage was from 1939 to 1954, called the initial stage of the development of recultivation legislation. The duration of the second stage is from 1955 to 1975: this was the main period, when the basis for recultivation legislation was created. The third stage in Russia (1976–1991) differed from in the USA and Europe (1971–present) in time but not in content; this stage was devoted to the development of organizational and economic infrastructure. The Russian case study shows four stages, the last one indicating a surge of interest in the recultivation issue. Considering the evolutionary changes in feasibility studies on recultivation, one of the most important features of feasibility studies on recultivation is the gradual complication over time by calculating the amount of income received including preventable economic damage. The main method of recultivation treatment was agricultural. The authors underline the link between calculating the economic efficiency of recultivation treatment and the existing ecological and economic paradigm. If in the initial stages the overall effect of recultivation effect was calculated by the income from agricultural products obtained on recultivated land, then the latest guidelines suggested, in addition to economic accounting, the inclusion of social and ecological outcomes. Currently, the environmental priorities of the economy are triggering the usage of the ecosystem approach for assessing the ecological result of recultivation. According to the results of our express analysis, the calculation of the "regulation of soil erosion" ecoservice in the context of different climatic zones has usually been done by employing the following methods: (1) market price (64%); (2) willingness to pay (27%); and (3) quantitative evaluation (9%). The most "expensive" soils to restore are those of the temperate and subtropical zones, which must be considered during the development of environmental management projects within the boundaries of these climatic zones.

Future research will focus on the development of a theory and a consistent approach to the social value-based assessment of ecosystems and changes in this assessment of ecosystems under the influence of anthropogenic impacts including land disturbance. This will be done by improving the concept of total economic value (TEV) and the collected database of assessments of ecosystem services in the context of different climatic zones in order to develop the universal methodological tools (guidelines) for assessing ecosystem services for substantiating the methods of recultivation. The choice of recultivation options (close to revitalization) would be based on, firstly, the criterion for

assessing the social value of alternative ecosystems planned for creation, and secondly, a new approach to ecological engineering, the fundamental difference in which is the creation of the most sustainable and maximally beneficial new ecosystem for man and nature.

**Author Contributions:** All authors contributed to the development of the current paper. The concept and methodology were prepared by M.I. and V.Y., while N.P. developed the legal framework governing the restoration of disturbed land. The detailed literature review, discussion and recommendations on the concept of recultivation and on the feasibility study on recultivation were done by both M.I. and V.Y. While M.I. examined the Russian vision, V.Y. researched the foreign experience and observed the economic evaluation of the "regelation of soil erosion" ecosystem service in the context of different climatic zones. The revision (review and editing) of the proposed literature review, discussion and recommendations were carried out by M.I., V.Y. and N.P. All authors have read and agreed to the published version of the manuscript.

**Funding:** The research was supported by the Ministry of Science and Higher Education in accordance with the state assignment for Ural State Mining University No. 0833-2020-0008, "Development and environmental and economic substantiation of the technology for reclamation of land disturbed by the mining and metallurgical complex based on reclamation materials and fertilizers of a new type". We obtained the scientific results with the staff of Center for the collective use by using the equipment of the Center for the collective use of scientific equipment of the Federal Scientific Center of biological systems and agricultural technologies of RAS (No Ross RU.0001.21 PF59, the Unified Russian Register of Centers for Collective Use (http://www.ckp-rf.ru/ckp/77384).

**Acknowledgments:** Emelyanova Evgenia Andreevna and Kostromina Tatyana Alekseevna, 3rd-year students at the Ural State Mining University, took part in collecting data and compiling the table on the economic evaluation of the "regulation of soil erosion" ecosystem service in the context of different climatic zones. The authors also thank the reviewers for valuable comments and suggestions.

**Conflicts of Interest:** The authors declare no conflicts of interest.

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
