# Peer review of "Recultivation of Post-Mining Disturbed Land: Review of Content and Comparative Law and Feasibility Study"

_resources, doi:10.3390/resources9060073_

Round 1

Reviewer 1 Report

This paper analyses the evolution of the conceptualization and definition of recultivation of disturbed lands. The structure of the study is consistent and the analysis is based on a large number of publications. The scientific contribution is relevant and provides the basis to future investigations.

I recommend the following minor corrections:

  • As this paper is based on the study of a large set of previous publications, I suggest to add a meta-analysis in the “materials and methods” sections, resuming the features of the publications considered in the study (typology, period, scope etc.).
  • Please avoid repetitions, e.g. the content of lines 123-130 and 424-428 was already reported previously in the paper.
  • English should be checked in all the manuscript.
  • In the conclusion, lines 440-441, it could be good to add one or two sentences recapping which were the three/four stages of the recultivation’s legal support evolution.

Author Response

Dear reviewer,

First of all, we want to thank you for your valuable comments and suggestions.

We highlighted all changes with green color in order to avoid lines numbering disadvantages. We put the lines numbers from the printed out paper in this cover letter. For some reasons, the numbers differ from the electronic version.

We have taken into consideration all your suggestions, such as:

  1. As this paper is based on the study of a large set of previous publications, I suggest to add a meta-analysis in the “materials and methods” sections, resuming the features of the publications considered in the study (typology, period, scope etc.)’ – We add the sentence ‘The meta-analysis has been done, which indicates the significant differences in the post-mining development understanding in different countries, caused by many aspects, such as: landscapes’ ecological engineering, responsibility and property rights’ issues, as well as land rarity criteria’ (lines 167-170).
  2. Please avoid repetitions, e.g. the content of lines 123-130 and 424-428 was already reported previously in the paper’ – We exclude the repetition (line 565)
  3. English should be checked in all the manuscript’. – The colleague of mine from the GB kindly agreed to edit the text. We attached the revised version of the paper.
  4. In the conclusion, lines 440-441, it could be good to add one or two sentences recapping which were the three/four stages of the recultivation’s legal support evolution’ – We add a couple of sentences devoted to the content of the stages of the recultivation’s legal support evolution (lines 598-603): ‘The first stage was from 1939 to 1954, it is called initial stage of recultivation legistlation’s development. The duration of the second one is from 1955 to 1975, this is the main period, when the recultivation legislation’s base was created. Russian third stage (1976-1991) differs from USA&Europe stage (1971-present) in time but not in the content; this stage is devoted to the organizational and economic infrastructure’s development. Russia case-study shows four stages, the last one indicates the surge of interest to recultivation issue.’

Best regards,

authors of the paper.

Reviewer 2 Report

  1. The experience of German researchers is not sufficiently discussed by the authors. In recent years, more than 30 coal mines have been recultivated in Germany.
  2. Line 97 “Employing the systemic and evolutionary approach based on at least a hundred of papers to the task, the clarification of recultivation’s concept has been done”. What hundred of papers do you mean? Put appropriate citations.
  3. It is not always necessary to talk about reclamation as a geoaesthetic. This term is good when reclamation is carried out in regions with a high concentration of population, industrial and civil facilities. In some cases it is impractical to engage in geoĐ°esthetics in remote areas of the country.
  4. Lines 224-225 “Unfortunately, it was the time-delayed transition to the market economy for Russia, which made the recultivation issue to become secondary for decades”. Very controversial. I am not sure that the market economy contributes to solving the problems of reclamation. And what does the phrase «time-delayed transition» to a market economy mean? In Soviet times, environmental problems, including problems of recultivation, were solved seriously. In particular, this is confirmed by the review of methods conducted by the authors of this paper.
  5. Lines 227-229. I think the authors have formulated this statement incorrectly, not quite accurately. The interest in recultivation cannot be associated with the spread of new economic models or the paradigm of the "green economy" ("circular economy"). The solution to the problems of reclamation, as well as other environmental problems, can only be connected with the environmental policy of the state, the development of environmental legislation and, as a result, increasing the social and environmental responsibility of mining companies. And after that, new economic models may already appear.
  6. The text is full of complex figures (data) for the perception of modern readers, such as the cost of 1 ha in the USSR in rubles and other monetary indicators of the Soviet Union, which has not existed for almost 30 years. The authors should also carefully explain such terms as, for example, the Standard of economic efficiency for the national economy.

  7. The paper doesn't look like a comprehensive research.

  8. The results and discussions section continue with a literary review, what is not correct.

  9. The connection of the considered methods (Line 251 -356)  with land recultivation after mining activities is unclear.

  10. How to check the validity of the results shown in figure?

  11. The results of the study presented in table 1 are unclear. Is this calculated by the authors?

Author Response

Dear reviewer,

First of all, we want to thank you for your valuable comments and suggestions.

We highlighted all changes with blue color in order to avoid lines numbering disadvantages.  We put the lines numbers from the printed out paper in this cover letter. For some reasons, the numbers differ from the electronic version.

We have taken into consideration all your suggestions, such as:

  1. The experience of German researchers is not sufficiently discussed by the authors. In recent years, more than 30 coal mines have been recultivated in Germany.’ – We look at German experience more thoroughly and add these scientific results to the text (we highlight these additions with yellow color (lines 161-162 ‘It is quite interesting that the origins of the idea of sustainability, its German expression of Nachhaltigkeit, lie in the 18th century [32,33].’; lines 177-178 ‘However, the first attempts to restore disturbed lands in Germany dated back to the end of the 19th century’; lines 231-232 ‘In 1940, Germany passed ‘The directives for the open spaces’ restoration caused by open cast mining’; lines 263-268 ‘In Germany in the Ruhr basin, since 1956 recultivation had been carried out according to plans that were developed simultaneously with mining plans. After the end of the Second World War in German Federal Republic, laws were adopted in the most regions to protect landscapes from destruction caused by open-cast mining. In addition to the technical aspect of nature conservation (disturbed lands’ recultivation), much attention was also paid to the conservation aspect (preservation of undisturbed landscapes) [40]’; lines 287-296 ‘However, German case study shows that this country still has problems with the mechanism and infrastructure [58]. For instance, Mansfeld district, which is associated with a more than 1000 year-old copper-mining tradition, was included in the German federal programme “Ökologische Großprojekte” (Major Ecological Projects) for the reculivation treatment [59]. Another quite a similar example is coal-mining-region “Zwickau-Lugau-Oelsnitz”. However, ‘one of the most crucial problems for the Zwickau-Lugau-Oelsnitz is the lack of outside funding for the new phase of rehabilitation’ [60]. These examples indicate serious obstacles in economic mechanism of recultivation treatment in Germany. As far as both infrastructure and economic mechanism are conserned, S. Krøijer and M. Kollöffel demonstrate that protests have been raised around the mining issue in the country in the research [61]’; lines 507-515 ‘As far as the last papers are concerned, for instance, the one of the German research [83] aimed at biological type of feasibility study, where measurement results were compared with previous data of microbial communities in sandy substrates and their impact to the environment’; line 517 ‘Other observed research concentrates on the ecosystem services’ type [86-88]’).  
  2. ‘Line 97 “Employing the systemic and evolutionary approach based on at least a hundred of papers to the task, the clarification of recultivation’s concept has been done”. What hundred of papers do you mean? Put appropriate citations’. – We mentioned all references here because all highlight the content of recultivation treatment from different directions (line 136).
  3. ‘It is not always necessary to talk about reclamation as a geoaesthetic. This term is good when reclamation is carried out in regions with a high concentration of population, industrial and civil facilities. In some cases it is impractical to engage in geoĐ°esthetics in remote areas of the country’ - We almost agree with your comment and make the changes according it (lines 136-139: ‘Authors prove that recultivation should be considered from the perspective of geoaesthetics in the regions with high concentration of population. It implies a harmonious incorporation of the recultivated landscape into the environment’).
  4. Lines 224-225 “Unfortunately, it was the time-delayed transition to the market economy for Russia, which made the recultivation issue to become secondary for decades”. Very controversial. I am not sure that the market economy contributes to solving the problems of reclamation. And what does the phrase «time-delayed transition» to a market economy mean? In Soviet times, environmental problems, including problems of recultivation, were solved seriously. In particular, this is confirmed by the review of methods conducted by the authors of this paper’ – We agree that we formulated the sentence incorrectly. Thank you very much that you detected this mistake and we can make our research better. All changes have been done in this case. We replace the ‘time-delayed’ with ‘time-consuming’ (lines 269-298: ‘Looking at Russian tradition, unfortunately, it was the time-consuming transition to the market economy for Russia, which made the recultivation issue to become secondary for decades’). In this sentence, we want to say that all development of the recultivation institute stopped at that time. Russian government had even more serious problems with economic and social policy during the 1990th and absolutely forgot about ecological problem, to which the recultivation has the direct link.
  5. Lines 227-229. I think the authors have formulated this statement incorrectly, not quite accurately. The interest in recultivation cannot be associated with the spread of new economic models or the paradigm of the "green economy" ("circular economy"). The solution to the problems of reclamation, as well as other environmental problems, can only be connected with the environmental policy of the state, the development of environmental legislation and, as a result, increasing the social and environmental responsibility of mining companies. And after that, new economic models may already appear’ – We absolutely agree with your comment. All changes have been done (figure 1 and lines 300-303: ‘This surge is linked to the continuous ecological deterioration of environment and the appearance of governmental management aimed to prevent these ecological obstacles. It trigged the spread of new economic models/paradigms, such as ‘green economy’ and circular economy that focused primarily on resource conservation’).
  6. The text is full of complex figures (data) for the perception of modern readers, such as the cost of 1 ha in the USSR in rubles and other monetary indicators of the Soviet Union, which has not existed for almost 30 years. The authors should also carefully explain such terms as, for example, the Standard of economic efficiency for the national economy’ – To meet the first suggestion, we add the information about currency exchange rate during the 60s to the 90s (lines 358-359: ‘The value of 1 ruble (RUB) varied from 0.6 to 1.85 USD during the 60s to the 90s’); to meet the second one – we add the sentence with the explanation in brackets (lines 399-400 ‘The standard of economic efficiency (profit margin) for the national economy during this period was 0.12’).
  7. ‘The paper doesn't look like a comprehensive research’ – Actually, we do not know what to say to this comment… We looked at different papers and databases all over the world in order to make a comprehensive research. It is impossible to investigate all world treasure to this task, but we tried our best to analyze information that we were able to get.
  8. ‘The results and discussions section continue with a literary review, what is not correct’ – We chose the review type of the manuscript. Maybe we are not right here, but according to PRISMA guidelines, it is not prohibited.
  9. The connection of the considered methods (Line 251 -356)  with land recultivation after mining activities is unclear’ – We try to clarify the connection in the certain paragraph (lines 499-521 ‘Summing up Russia case study, we can identify five types of feasibility study’s main criteria of recultivation treatment, such as 1) benefits type; 2) standard type; 3) cost type; 4) biological type and 5) ecosystem services’ type. All these types can be used both at the project step of recultivation treatment and at the result step, when recultivation has been done, in order to evaluate economic efficiency of the treatment. According to scientometric analysis [82], the USA, England, Canada, Germany, China, Italy, and Czech Republic focus on recultivation issues and published lots of articles because of land and environment policies. Moreover, ‘sustainable regeneration, urban brownfields’ regeneration, mental distribution, coal-mine brownfield, and ecosystem service were the identified co-citation clusters and represented the hot topics and emerging trends’ [82]. As far as the last papers are concerned, for instance, the one of the German research [83] aimed at biological type of feasibility study, where measurement results were compared with previous data of microbial communities in sandy substrates and their impact to the environment. The same type of feasibility study’s main criteria has been chosen by Z. Zhang et al [84]. Costs-benefits types are discussed in the paper [85]. Other observed research concentrates on the ecosystem services’ type [86-88], etc. Therefore, the review on feasibility study of recultivation treatment shows the emerging and growing trend of ecosystem services’ type during the last years. The main question arises: ‘How to calculate the ecosystem sevices in order to evaluate the economic efficiency of recultivation treatment?’).
  10. ‘How to check the validity of the results shown in figure?’ – If we understand correctly, the reviewer meant the figure 2. The database to this figure is the Table 1. We add this clarification in the text (line 539-540 ‘The numbers have been calculated by employing the data presented in the Table 1’).
  11. ‘The results of the study presented in table 1 are unclear. Is this calculated by the authors?’ – The assessments was obtained by the authors of the papers [93-102]. We just recalculated the numbers in order to make it comparable by using the discounting tool for identification the differences in value caused by climatic zone. We add it clarification too (lines 541-543 ‘Economic assessments obtained in the papers [93-102] have been recalculated by the authors of this article up to the date 01/01/2020 by using the discounting tool’). We have done all calculation in MS Excel, we can send it to reviewer or even attached it to the manuscript.

Best regards,

authors of the paper.

Round 2

Reviewer 2 Report

Dear authors,

I think the you did a good job on the comments in the paper.

I am satisfied with your answers.